# Immunopharmacological Activities of Luteolin in Chronic Diseases

**DOI:** 10.3390/ijms24032136

**Published:** 2023-01-21

**Authors:** Lei Huang, Mi-Yeon Kim, Jae Youl Cho

**Affiliations:** 1Department of Biocosmetics, Sungkyunkwan University, Suwon 16419, Republic of Korea; 2School of Systems Biomedical Science, Soongsil University, Seoul 06978, Republic of Korea

**Keywords:** luteolin, flavonoid, chronic disease, cancer, bioinformatics analysis, in vivo, in vitro

## Abstract

Flavonoids have been shown to have anti-oxidative effects, as well as other health benefits (e.g., anti-inflammatory and anti-tumor functions). Luteolin (3′, 4′, 5,7-tetrahydroxyflavone) is a flavonoid found in vegetables, fruits, flowers, and herbs, including celery, broccoli, green pepper, navel oranges, dandelion, peppermint, and rosemary. Luteolin has multiple useful effects, especially in regulating inflammation-related symptoms and diseases. In this paper, we summarize the studies about the immunopharmacological activity of luteolin on anti-inflammatory, anti-cardiovascular, anti-cancerous, and anti-neurodegenerative diseases published since 2018 and available in PubMed or Google Scholar. In this review, we also introduce some additional formulations of luteolin to improve its solubility and bioavailability.

## 1. Introduction

Flavonoids are in a wide variety of natural products [1,2]. The fundamental structure of flavonoids involves an aromatic A ring, a heterocyclic C ring, and an aromatic B ring that are joined by a carbon–carbon bridge. Flavonoids are divided into several subgroups based on their molecular backbone structure and hydroxyl group: flavones (2-phenylchromen-4-one), flavonols (3-hydroxy-2-phenylchromen-4-one), flavanones (2, 3-dihydro-2-phenylchromen-4-one), flavanols (2-phenyl-3,4-dihydro-2H-chromen(flavan)-3-ol, flavan-4-ol, flavan-3,4-diol), flavanonols (3-hydroxy-2, 3-dihydro-2-phenylchromen-4-one), isoflavones (3-phenylchromen-4-one), and anthocyanidins (2-phenylchromenylium) [3,4]. Flavonoids are widely found in tea, vegetables, fruits, and wine [5,6]. Because of their anticancer, anti-oxidative, anti-inflammatory, anti-mutagenic, anti-allergy, and cardioprotective capabilities, flavonoids are used in a variety of pharmaceutical, nutraceutical, medicinal, and cosmetic applications [7].

One of the most common flavones is luteolin (3′, 4′, 5,7-tetrahydroxyflavone), which has a yellow crystalline appearance [8]. The chemical structure of luteolin is shown in Figure 1, and its molecular formula is C_15_H_10_O_6_. Many vegetables, medicinal herbs, and fruits are rich in luteolin, including carrots, broccoli, cabbages, parsley, thyme, peppermint, basil, celery, artichoke, and apples [9]. The active metabolite or derivatives of luteolin such as luteolin glucuronide and luteolin 7-glucoside have been reported to have anti-oxidant, anti-tumor, anti-microbial, anti-inflammatory, anti-apoptotic, anti-allergy, anti-diabetic, chemotherapeutic, cardioprotective, and neuroprotective properties [10]. Most studies of luteolin have focused on its anti-inflammatory effects.

Inflammation is part of the sophisticated biological response of body tissues and cells to pathogens and noxious physical or chemical stimuli. Inflammation helps to eliminate the initial cause of cellular damage, remove necrotic cells and tissue damaged during the original insult and the inflammatory process, and initiate tissue repair [11]. Its five main signs are heat, pain, redness, swelling, and loss of function. Inflammation can be acute or chronic. As the body’s first response to harmful stimuli, acute inflammation increases the movement of plasma and leukocytes (especially granulocytes) from the blood to the injured tissue. The inflammatory response is propagated and matured by a series of biochemical events that involve the local vascular system, the immune system, and various cells within the injured tissue [12,13]. Chronic inflammation, in contrast, is a protracted, dysregulated, and maladaptive response that causes tissue damage, active inflammation, and attempts at tissue repair [14,15]. Numerous chronic human diseases, such as allergies, atherosclerosis, cancer, arthritis, and autoimmune disorders, are linked to chronic inflammation. The processes involved in the occurrence and development of acute inflammation are well-defined, but the causes and molecular and cellular pathways of chronic inflammation are poorly understood [16]. The biomarkers of inflammation, including cytokines such as tumor necrosis factor-α (TNF-α), interleukin-1 (IL-1), IL-6, IL-8, and monocyte chemotactic protein 1 and other enzymes and proteins such as cyclooxygenase-2 (COX-2) and the matrix metalloproteinases (MMPs) [17,18,19] are becoming increasingly important in the study of various diseases.

In this review, we discuss the pharmacological activities of luteolin in vitro and in vivo (Figure 2). Our goal is to provide a clear direction for future luteolin research.

Many diseases cause inflammation in the body’s tissues and organs. Like other flavonoids, luteolin exhibits a variety of pleiotropic properties, making it difficult to attribute all of its pharmacological effects to a single metabolic process. Luteolin has anti-inflammatory properties at micromolar concentrations, including the restraint of proinflammatory mediators (e.g., COX-2, nitric oxide (NO), IL-6, IL-1, and TNF-α) and the regulation of several signaling pathways, including the nuclear factor (NF)-κB, activator protein (AP)-1, and JAK–STAT pathways. Crosstalk connects all those pathways, and luteolin can control and repress them. The most important effect of luteolin, however, is its potent antioxidative power, which includes excellent radical scavenging and cytoprotective properties [9,20,21].

In the following sections, we focus on the immunopharmacological effects of luteolin in asthma, renal damage, cardiovascular diseases (myocardial infarction and atherosclerosis), tumors (lung cancer, hepatocellular carcinoma, colorectal cancer, and pancreatic cancer), and neurodegenerative diseases (Alzheimer’s disease and Parkinson’s disease) during the past five years. In addition, bioinformatics analysis was performed to identify differentially expressed genes (DEGs) of cytokines/chemokines under luteolin-treated conditions from a publicly available dataset.

## 2. Anti-Inflammatory Activity of Luteolin

### 2.1. Gastritis

A histological definition of gastritis is mucosal inflammation, but infections are what lead to acute gastritis. The two most significant types of chronic gastritis are inflammation caused by *Helicobacter pylori* and metaplastic atrophic gastritis, which has an autoimmune origin [22]. Radziejewska et al. found that in the CRL-1739 *H. pylori*-infected gastric cancer cell line, luteolin changed the expression of the MUC1 extracellular domain, sT antigen, ADAM-17, IL-8, IL-10, and NF-κB. Those experiments left many unanswered questions. For instance, in subsequent studies, it will be important to determine how *H. pylori* influences cell proliferation in multiple infections [23]. Several studies on lipopolysaccharide (LPS)-treated RAW264.7 cells and mice with HCl/EtOH-triggered gastritis reported that natural herbs targeted Src and Syk in the NF-κB pathway and had anti-inflammatory effects by blocking NF-κB signaling. Luteolin is one of the main active ingredients in those natural herbs [24,25].

### 2.2. Arthritis

Arthritis occurs in more than 100 different forms, the most prevalent of which are rheumatoid arthritis, osteoarthritis (OA), psoriatic arthritis, and inflammatory arthritis [26]. Xue et al. found that luteolin efficiently reduced NO, TNF-α, and IL-6 levels, as well as the expression of JNK and p38MAPK in OA cartilage cells. Luteolin also successfully suppressed the growth of OA cartilage cells [27]. Zhou et al. demonstrated the therapeutic effect of luteolin on OA using both in vivo and in vitro experiments. According to their in vitro research, luteolin prevented H_2_O_2_-induced cell death, apoptosis, oxidative stress, programmed necrosis, and the production of inflammatory mediators in primary murine chondrocytes. Additionally, AMPK functions as a favorable upstream regulator of Nrf2 and could activate the Nrf2 and AMPK pathways when luteolin is present. The results from their in vivo experiments showed that luteolin treated OA in the destabilization of the medial meniscus mouse model [28]. Fei et al. found that luteolin inhibited inflammatory factors, including TNF-α, inducible nitric oxide synthase (iNOS), COX-2, MMP-1, MMP-2, MMP-3, MMP-8, MMP-9, MMP-13, MMP upregulation, and collagen II degradation in IL-1β-induced chondrocytes [20].

### 2.3. Asthma

Asthma is a common chronic inflammatory disorder of the lungs caused by heterogenic gene–environment interactions that are not fully understood. It is characterized by airway obstruction and bronchial hyperresponsiveness. Clinically, asthma is distinguished by recurrent episodes of wheezing, coughing, chest tightness, and shortness of breath [29,30]. The results of studies in the past five years have shown that luteolin has significant therapeutic effects on asthma. Regulatory T (Treg) cells are important in autoimmunity and have been shown to reduce inflammation in allergic asthma [31,32]. In 2018, Kim et al. confirmed that luteolin could effectively inhibit airway inflammation and T helper 2 (Th2) upregulation in ovalbumin antigen (OVA)-sensitized mice. They used BALB/c mice sensitized to OVA via aerosol challenge to establish an airway inflammation model. Luteolin significantly reduced the levels of IgE, IL-4, IL-5, and IL-13 in bronchoalveolar lavage (BAL) fluid, as well as CD19^+^ B cells, CD4^+^ T cells, CD3^-^CCR3^+^ cells, and CD3e^+^Gr-1^+^ cells in lung tissue. Conversely, the level of IFN-γ, a Th1 cytokine, was increased following luteolin treatment. In vitro, the researchers demonstrated that CD4^+^CD25^+^foxp3^+^ (iTreg) cells have functional Treg cell activity. After the luteolin-induced adoptive transfer of these cells, the levels of TGF-β1 and foxp3 mRNA expression were elevated in lung tissue. Following an allergen challenge, iTreg cells that were transferred into OVA-sensitized mice reduced airway hyperresponsiveness, eosinophil recruitment, and eotaxin, IgE, and Th2 cytokine expression and increased IFN-γ production in BAL fluid. Additionally, an asthma mouse model in which CD25 was reduced was protected from illness via the adoptive transfer of iTreg cells. Luteolin could offer a fresh approach to asthma management by stimulating foxp3 and CD4^+^CD25^+^ Treg cells [33]. Autophagy is a highly conserved biological mechanism for breaking down excess and dysfunctional components within cells, such as misfolded/aggregated proteins, damaged organelles, and invading pathogens, so it plays an important physiological role in maintaining cellular homeostasis. However, factors such as oxidative stress and inflammation caused by excessive autophagy can adversely affect cells and even lead to cell death [34,35]. Wang et al. wondered whether the inhibition of excessive autophagy could significantly alleviate asthma symptoms. They made an allergic asthmatic mouse model using OVA and found through ELISAs, histological observation, Western blotting assays, and immunohistochemistry that luteolin inhibited autophagy in the lung tissues of asthmatic mice by activating the PI3K/Akt/mTOR signaling pathway and restraining the beclin1-PI3KC3 protein complex [36]. The severe acute respiratory syndrome coronavirus 2 (SARS-CoV-2) pandemic, which started in December 2019, is a worldwide health emergency [37]. In 2020, one study suggested that adults with asthma are at higher risk for severe COVID-19, driven by an increased risk among those with nonallergic asthma. In contrast, patients with allergic asthma did not have a significantly increased risk [38]. System pharmacology and bioinformatics analyses were used by Xie et al. to identify the top six core targets of luteolin against the COVID-19/asthma comorbidity and identified TP53, AKT1, ALB, IL-6, TNF, and VEGFA. Their findings suggested that luteolin might regulate some signaling pathways to prevent the COVID-19/asthma comorbidity. They also provided guidance for future study by highlighting the practical and important pharmacological targets of luteolin in addressing the COVID-19/asthma comorbidity, but their anticipated outcomes still require thorough verification [39]. Macrophages are cells that respond to inflammation by polarizing into either traditionally activated macrophages (M1) or alternatively activated macrophages (M2). Asthma is worsened by inflammation, which also causes the release of Th2 cytokines, including IL-4 and IL-13, that subsequently stimulate the development of M2 macrophage markers [40,41]. Many M1 macrophages are found in asthma patients, and reducing M1 macrophages and increasing M2 macrophages is one way to treat asthma [42]. In 2022, Gong et al. demonstrated that luteolin increased M2 polarization and decreased M1 polarization in macrophages that had been stimulated by THP-1. However, because that research was only conducted in vitro, further testing in live animal models, as well as human cell or tissue samples, is needed [43]. Some researchers have used network pharmacology, molecular docking, and bioinformatics to elucidate some traditional drugs that are commonly used but whose pharmacological mechanisms of action are unknown, including Chuankezhi injection, ephedra, modified Guo Min decoction, Gerbera Piloselloides Herba, Qi-Wei-Du-Qi-Wan, Huanglong cough oral liquid, *Hyssopus cuspidatus Boriss*, conciliatory anti-allergic decoction, *Cucumis sativus L.*, *Ziziphus mauritiana Lam* leaves, Zhisou San, and *Vochysia divergens Pohl* (popularly known as “Cambará”) [44,45,46,47]. Together, those studies demonstrated that luteolin could reduce airway inflammation and increase Th2 in OVA-sensitized mice while also effectively reducing autophagy levels in the lungs of asthmatic mice. The therapeutic potential of luteolin was also shown to be excellent for the COVID-19/asthma comorbidity. By increasing the expression of hsa circ 0001326, luteolin enhanced M2 polarization and inhibited M1 polarization. Luteolin also mediated the expression of hsa-miR-136-5p and USP4. These discoveries provided fresh perspectives on the management of asthma.

### 2.4. Renal Damage

Because renal damage is a common and serious disease, it is necessary to find a way to treat it. Researchers used inorganic mercury (HgCl_2_) to induce renal damage in male Wistar rats. In one such study, luteolin was found to attenuate HgCl_2_-induced renal damage by activating the AMPK/mTOR autophagy pathway. In another study, researchers found that luteolin treatment increased Nrf2 nuclear translocation and enhanced nicotinamide adenine dinucleotide phosphatase:quinone-acceptor 1 (NQO1) and heme oxygenase-1 (HO-1) expression. Luteolin was found to play a considerable role in the activation of Nrf2, which attenuated HgCl_2_-induced renal toxicity through antioxidant, anti-apoptotic, and anti-inflammatory actions. Thus, luteolin could be a potential drug to prevent HgCl_2_-induced renal damage [48,49]. Moreover, luteolin attenuated angiotensin II-induced renal injury in apolipoprotein E-deficient mice (ApoE^−/−^). It also alleviated the liver and kidney dysfunction induced by doxorubicin [50,51].

### 2.5. Lung Injury

Some researchers have investigated whether luteolin helps to reduce lung injury in vivo using the mouse cecal ligation and puncture (CLP) model. Luteolin’s powerful anti-inflammatory properties, in the form of triggering the activity of Tregs, were demonstrated by Zhang et al. The protein levels of caspase-11, caspase-1, gasdermin D (GSDMD), IL-1α, and IL-1β in the lungs of CLP mice treated with luteolin were significantly decreased compared with the control [52]. Xie et al. also reported that luteolin attenuated lung injury and inhibited uncontrolled inflammation by inducing the differentiation of CD4^+^ CD25^+^ FOXP3^+^ Tregs and upregulating IL-10 expression. In addition, the anti-inflammatory cytokine IL-10 promoted the polarization of M2 macrophages in vitro. Luteolin-induced Treg differentiation from naive CD4^+^ T cells could be a mechanism regulating IL-10 production [53]. Luteolin decreased glutathione levels and superoxide dismutase activity in rat lung tissue, and a significant reduction in malondialdehyde was also found after luteolin treatment. Luteolin administration decreased the expression of TNF-α and IL-10 mRNA in lung tissue. Moreover, luteolin was shown to attenuate sepsis-induced acute lung injury in mice by inhibiting intercellular adhesion molecule (ICAM)-1, NF-κB, oxidative stress, and part of the iNOS pathway [54,55]. Luteolin could be an effective candidate to treat HgCl_2_-induced lung injury by blocking NF-κB activation and activating the AKT/Nrf2 pathway. Luteolin attenuates LPS-induced bronchopneumonia injury by downregulating miR-132 in vitro and in vivo. These findings provided a theoretical basis for the further exploration of luteolin in the treatment of bronchopneumonia in children. Luteolin inhibited the activity of proinflammatory mediators (HMGB1, iNOS/NO, COX-2, and NF-κB) in RAW264.7 cells activated by LPS. Luteolin has great potential in the treatment of acute lung injury (ALI) and related diseases for which HMGB1 is a therapeutic target [56,57,58]. In conclusion, luteolin can improve lung injury induced by HgCl_2_, LPS, and CLP.

## 3. Anti-Cardiovascular Disease Activity of Luteolin

Heart and blood vessel problems are collectively referred to as cardiovascular diseases (CVDs) by the World Health Organization. CVDs are the greatest cause of death globally [59,60]. Many studies have shown that luteolin is crucial in the treatment of CVDs. In this investigation, we focus on myocardial infarction (MI) and atherosclerosis (AS).

### 3.1. Myocardial Infarction

MI is pathologically defined as the death of cardiomyocytes due to prolonged ischemia. MI is an acute heart disease characterized by the reduced or complete cessation of blood flow to a portion of the heart muscle, resulting in an imbalance in the supply and delivery of oxygen to the heart muscle that, in turn, leads to cardiac cell death [61]. In 2020, Zhao et al. randomly divided forty healthy rats into four groups and measured their left ventricular-related data, creatine kinase isoenzymes, lactate dehydrogenase, and myocardial pathological data, including inflammatory cytokines in the serum and heart. The levels of TNF-α, IL-6, and IL-1β in the sera and hearts of the ischemia/reperfusion (I/R) group were higher than those in the sham operation group, whereas the levels in the I/R and Lu group were significantly lower than those in the I/R rats. In the hearts of the I/R and Lu group, Sirt1 levels were higher than those in the I/R group, whereas NLRP3, ASC, caspase-1, IL-1β, p-NF-κB/p65, and p-IκBα levels were lower. They found that luteolin had a protective effect on rats’ I/R, and the Siti1/NLRP3/NF-κB pathway played a key role in that. To offer fresh insights for into the clinical management of I/R reperfusion damage, more research into its intrinsic mechanism is necessary [62,63]. In another study, researchers pretreated H9c2 cells with different concentrations of luteolin and then stimulated them with LPS or left them unstimulated. LPS stimulated the mRNA and protein expression of MCP-1, TNF-α, IL-6, IL-1β, and IL-18 in H9c2 cells, and their expression levels decreased as the luteolin concentration increased. To further determine whether NLRP3 is involved in the protection from oxidative stress offered to H9c2 cells by luteolin, NLRP3 mRNA expression was decreased following transfection with NLRP3 siRNA. The authors found that NLRP3 knockdown enhanced the viability of LPS-treated H9c2 cells and reduced LPS-induced apoptosis and inflammatory factors. Luteolin has been suggested to reduce inflammatory damage by downregulating NLRP3. It was found that luteolin could protect cardiomyocytes from LPS-induced apoptosis and inflammatory damage. These findings support the use of luteolin in the treatment of myocarditis and support NLRP3 as a potential therapeutic target for myocarditis [64]. The global epidemic of diabetes is increasing the burden of CVD, which is the leading cause of death among people with diabetes [65]. In particular, diabetes causes specific cardiac functional and structural abnormalities that lead to diabetic cardiomyopathy (DCM) [66]. In 2019, Li et al. induced DCM in C57BL/6 mice using an intraperitoneal injection of streptozotocin, and they induced H9c2 cell injury with high glucose in vitro. Then, they studied cardiac fibrosis, hypertrophy, inflammation, and oxidative stress in those in vitro and in vivo models. They found that luteolin decreased reactive oxygen species (ROS) production and increased the amount of Nrf2 and HO-1. Thus, luteolin might maintain heart function by reducing the oxidative damage and inflammation caused by diabetes [67]. In 2020, Hu et al. proposed that luteolin attenuated myocardial injury, improved left ventricular function, reduced cardiomyocyte apoptosis, and enhanced SERCA2a transcriptional activity by upregulating Sp1 expression. In that way, luteolin provided protection against myocardial I/R damage in mice, indicating that Sp1 regulation might be a crucial anti-apoptotic mechanism that enhances the contractile capacity of cardiomyocytes. The exact mechanism by which luteolin treatment shields myocardium from I/R damage is not yet fully understood. However, these findings create the groundwork for applying luteolin treatment in cardiovascular therapy by offering a fresh line of inquiry and an experimental basis for investigation [68]. By regulating the SHP-1/STAT3 signaling axis, luteolin produced cardiac protective effects in hypoxia/reoxygenation (H/R)-treated cells and I/R animals [69]. In 2020, Wu et al. randomly divided experimental mice into three groups (25 mice in each group): a control group, LPS group, and LPS and luteolin group. A reversible myocardial depression complication of severe sepsis and septic shock is called sepsis-induced cardiomyopathy (SIC). Sepsis is a condition that is frequently caused by an inappropriate host response to infection, and it can induce fatal organ malfunction [70]. Those authors used luteolin to reduce sepsis-induced cardiac damage in mice by increasing autophagy. Luteolin exerted its cardioprotective effect on SIC by reducing the levels of inflammatory cytokines such as IL-1β, IL-6, and TNF-α. In summary, the results of that investigation supported the notion that luteolin is a potential therapeutic agent for the management of SIC. Luteolin’s pharmacological actions, including its antioxidant and anti-inflammatory qualities, could contribute to its anti-SIC benefits. However, that study had severe flaws. The cardioprotective benefits of luteolin against SIC could also be linked to the activation of AMPK signaling, which reduces apoptosis and increases autophagy. Those results could provide a foundation for a promising and groundbreaking therapeutic approach for SIC [70,71]. Due to its capacity to control the inflammatory alterations brought on by high-fat diet (HFD), luteolin was able to offset the decreased expression of miR-214-3p in mesenteric arteries from HFD mice [72]. The deleterious inflammatory consequences of a high-cholesterol and high-fat (HCHF) diet can be prevented by luteolin. The cardioprotective effect of luteolin might be achieved via reductions in the two main pro-inflammatory cytokines, TNF-α and IL-18, which are linked to the heart dysfunction associated with obesity, as well as via its antioxidant effects [73].

### 3.2. Atherosclerosis

AS, considered to be a major cause of CVD, is a chronic vascular inflammatory disease associated with high morbidity and mortality. Inflammatory cytokines play an important role in the development and progression of AS. AS development and progression are also linked to the abnormal lipid metabolism and inflammatory infiltration of the arterial wall [74]. The proliferation and migration of vascular smooth muscle cells (VSMCs) are crucial in the development of arterial remodeling in a variety of vascular disorders, including AS, hypertension, and associated conditions. A particular TGF-β receptor called transforming growth factor-β receptor 1 (TGFBR1) has been identified as a key player in the atherosclerotic process. It mediates both the physiological and pathological effects of TGF-β. Wu et al. investigated whether TGFBR1 signaling underlies the inhibitory effect of luteolin on VSMC proliferation and migration. They demonstrated that luteolin inhibited VSMC proliferation and migration by inhibiting TGFBR1 signaling without inducing cell damage. The results of that study provide new insights into the anti-atherogenic mechanism of luteolin. However, that study had some limitations, and further studies are needed to investigate the molecular mechanism of luteolin in vivo and in vitro [75,76,77]. Ding et al. discovered that luteolin significantly reduced AS in ApoE^−/−^ mice fed a high-fat diet by reducing inflammation. They also discovered that by suppressing signal transducer and activator of transcription 3 (STAT3) in vitro, luteolin reduced oxidized, low-density lipoprotein-induced inflammation (including the mRNA production of ICAM-1, VCAM-1, TNF-α, and IL-6). They discovered that STAT3 could be a viable therapeutic target that might stop the progression of AS and that luteolin might be a good candidate drug for AS [78]. As previously stated, lipid metabolic failure and the inflammatory infiltration of the arterial wall are linked to the onset and progression of AS. The expression of the macrophage marker CD68, the macrophage chemokine CCL2, and the inflammatory cytokines IL-6 and TNF-α in the aortic root decreased after luteolin supplementation. Additionally, luteolin has been demonstrated to dose-dependently reduce the production of macrophage chemokines and inflammatory cytokines in ThP-1-derived macrophages [79].

## 4. Anti-Tumor Activity of Luteolin

Cancer is a major health problem worldwide and refers to a group of diseases caused by the abnormal growth of cells with invasive potential. Many environmental and genetic factors cause cancer in humans. In addition, oxidative stress plays an important role in the pathophysiology of different types of cancer. As a result, antioxidants have received a lot of attention as new therapeutic strategies for cancer. Luteolin has begun to attract the attention of researchers because of its ability to target oncogenic cell signaling pathways in different cancers. We next review the progress of luteolin research in the past five years in terms of the flavonoid’s anticancer effects and molecular mechanisms [80,81].

### 4.1. Lung Cancer

Lung cancer is a leading cause of cancer-related death worldwide. It is a complicated disease with numerous histological and molecular categories of clinical importance. As researchers in this era of molecularly targeted therapy work to target the molecular pathways involved in the etiology and pathogenesis of lung cancer, new medications are constantly being tested in preclinical and clinical trials [82,83].

Yu et al. reported that AIM2, pro-caspase-1, caspase-1 p10, pro-IL-1β, and IL-1β were inhibited by luteolin in vitro and in vivo [84]. Zhang et al. found that luteolin prevented lung cancer cells from proliferating and that it triggered cell cycle arrest and apoptosis. LIMK1 and associated signaling pathways are inhibited to mediate those actions [85]. In an in vivo study, Masraksa et al. reported that through the elimination of focal adhesion formation and the inhibition of FAK–Src signaling, luteolin therapy prevented non-small cell lung cancer (NSCLC) from migrating. Rac1, Cdc42, and RhoA, which control the actin cytoskeleton and cell migration, are similarly altered by luteolin [86]. In addition, Lin et al. revealed that luteolin inhibited the synthesis of pro-MMP-2 and ICAM-1, as well as the ability of H460 cells to migrate when fine particulate matter (PM2.5) was present. Luteolin also inhibited the PM2.5-induced EGFR–PI3K–AKT pathway [87]. In a combination study, Wu et al. reported that tumor necrosis factor-related apoptosis-inducing ligand (TRAIL) in combination with luteolin could be a successful NSCLC chemotherapeutic approach [88].

### 4.2. Hepatocellular Carcinoma

Hepatocellular carcinoma (HCC) is a common primary liver malignancy and a leading cause of cancer-related mortality worldwide. Its prognosis is poor, and its incidence is high [89]. Im et al. reported that luteolin can induce the apoptosis of human HCC SK-Hep-1 cells, which have high osteopontin (OPN) expression, by inhibiting the AKT/OPN pathway, and luteolin has been used to treat cancer with high OPN expression and found to have low toxicity [90]. More surprisingly, Lee et al. demonstrated that luteolin-induced endoplasmic reticulum (ER) stress might exert anticancer effects in a p53-independent manner. Luteolin only induced autophagy in Hep3B (p53 null type) cells, thereby enhancing cell viability [91]. In addition, Yang et al. reported that luteolin upregulated miR-6809-5p to inhibit the growth of HCC cells. The overexpression of miR-6809-5p could inhibit flotillin 1 expression in HCC cells and inactivate the Erk1/2, p38, JNK, and NF-κB/p65 signaling pathways, thus inhibiting growth and inducing apoptosis [92]. In a combination study, co-treatment with sorafenib (a small-molecule multi-kinase inhibitor) and luteolin killed HCC cells through JNK-mediated apoptosis. Thus, luteolin might be an ideal candidate to increase the activity of sorafenib in HCC treatment [93].

### 4.3. Colorectal Cancer

Colorectal cancer (CRC) is one of the most common gastrointestinal tumors. It is the third most recurrent cancer and the fourth most likely cause of cancer death worldwide, and its incidence has been increasing in recent years [94]. Several studies have found that luteolin plays a role in CRC. Kang et al. demonstrated that luteolin had anticancer effects on CRC cells by inducing apoptosis. That effect depended on increased Nrf2 transcription mediated by the induction of the DNA demethylation of its promoter. In addition, luteolin increased the interaction between Nrf2 and p53, which increased the expression of antioxidant enzymes and apoptotic proteins. Those findings provide insights into the potential use of luteolin in the prevention and treatment of cancer [95]. Research by Zuo et al. supported luteolin’s inhibitory effects on cell growth and colony formation in HCT116 cells. They also demonstrated that luteolin activated the mRNA and protein expression of Nrf2 and its downstream phase II detoxifying antioxidant enzymes HO-1 and NQO1. Most importantly, they demonstrated that luteolin can epigenetically regulate Nrf2 expression by inhibiting DNA methyltransferase (DNMT) and histone deacetylases (HDAC) activity [96]. In HCT116 colon cancer cells, luteolin induced apoptosis and autophagy through a p53-dependent pathway [97,98]. Song et al. showed that luteolin could prevent CRC cells from proliferating, stop the cell cycle, and cause DNA damage and the progression of apoptosis by acting through the MAPK pathway. According to their findings, luteolin has the potential to be an effective antineoplastic agent and future CRC treatment adjunct [99]. Luteolin dose-dependently decreased the viability and proliferation of metastatic human colon cancer SW620 cells and increased the expression of antioxidant enzymes. After luteolin treatment, the expression of the anti-apoptotic protein Bcl-2 decreased, and the expression of the pro-apoptotic protein Bax and caspase-3 increased. Luteolin has antitumor activity in metastatic human colon cancer SW620 cells through an ERK/forkhead box O3a (FOXO3a)-dependent mechanism, and it has antimetastatic potential [100]. Yao et al. found that luteolin inhibited the migration and invasion of CRC cells in vitro and in vivo, but it did not affect the proliferation of CRC cells. Further experiments revealed that luteolin inhibited CRC cell metastasis by regulating the miR-384/pleiotrophin axis [101]. Combined treatment with luteolin and oxaliplatin synergistically inhibited the growth of mouse HCT116 xenograft tumors by promoting apoptosis and inhibiting proliferation, most likely through an AMPK-related mechanism. Those findings suggest that a luteolin-rich diet could enhance the efficacy of oxaliplatin in CRC [102].

### 4.4. Pancreatic Cancer

Pancreatic cancer is a malignant tumor with poor prognosis and high mortality. The survival rate for pancreatic cancer has remained relatively unchanged since the 1960s despite significant improvements in treating other major forms of cancer [103]. Li et al. observed that luteolin triggers apoptosis in pancreatic cancer cells (SW1990 cells) by targeting BCL-2 and that it could be used as a potential medication to treat this cancer [104]. Kato et al. modeled N-nitroso-bis(2-oxopropyl)-amine-induced hamster pancreatic ductal adenocarcinoma (PDAC) and found that luteolin targets a novel STAT3–dihydropyrimidine dehydrogenase (DPYD) pathway, making it a promising chemo-prophylaxis agent for PDAC [105]. One study proposed that miRNA-301-3p was downregulated after treatment with luteolin, which showed an in vitro antiproliferative effect on PANC-1 (human pancreatic cancer) cells [106]. Huang et al. injected caerulein into six-week-old male C57BL6 mice to produce an acute pancreatitis model. Their results showed that luteolin inhibited tubular complex formation and the ectopic expression of cytokeratin-19. Luteolin also decreased SOX9, p-STAT3, and p-EGFR protein levels. These findings indicate that luteolin has potential anti-carcinogenic action on the pancreas, which is worthy of further study [107].

### 4.5. Other Cancers

Several studies have shown that luteolin plays a role in other cancers. For instance, breast cancer (BC) is the most common cancer in women worldwide, and triple-negative breast cancer (TNBC) accounts for 20% of all BC cases. Wu et al. showed that luteolin inhibited the proliferation and metastasis of androgen receptor-positive TNBC by inducing the AKT/mTOR signaling pathway to reduce the levels of H3K27Ac and H3K56Ac, thereby regulating the expression of *MMP-9*. Cao et al. reported that luteolin inhibited YAP/TAZ oncogenic activity in highly metastatic TNBC cells, suggesting that luteolin is a potential candidate for TNBC therapy [108,109]. Luteolin inhibited tamoxifen resistance in BC and can inhibit BC cell growth by targeting human telomerase reverse transcriptase. Luteolin is a promising natural drug to inhibit BC invasion and metastasis [110,111,112]. Periodontal disease is a chronic infectious disease of periodontal tissue. The results of Casili et al. demonstrated the anti-inflammatory properties of luteolin in LPS-induced periodontitis in rats. Yuce et al. found that luteolin prevented bone loss in experimental periodontitis. Luteolin significantly reduced alveolar bone loss by decreasing osteoclast activity and the expression of MMP-8 and receptor activator of NF-κB ligand, increasing osteoblast activity, and upregulating the expression of *tissue inhibitor of MMP-1*, *bone morphogenetic protein-2*, and *osteoprotegerin (OPG)*. *iNOS* expression also decreased with luteolin administration [113,114]. Luteolin has been shown to ameliorate diseases related to inflammation-induced retinal degeneration by inhibiting the NF-κB and MAPK pathways in IL-1β-stimulated ARPE19 cells [115,116].

## 5. Anti-Neurodegenerative Disease Activity of Luteolin

A group of chronic, progressive disorders known as neurodegenerative diseases affect specific regions of the brain, spinal cord, or peripheral nerves and are characterized by the gradual loss of neurons. As a result, cognition, movement, strength, coordination, sensation, or autonomic control are impaired. Neuroinflammation plays a major role in several neurodegenerative diseases, such as Alzheimer’s disease (AD), Parkinson’s disease (PD), and multiple sclerosis, and it is characterized by the activation of microglia and astrocytes, the secretion of pro-inflammatory cytokines and chemokines, and the recruitment of immune cells from the periphery. Here, we discuss the role of luteolin in AD and PD [117,118].

### 5.1. Alzheimer’s Disease

AD is a progressive neurodegenerative disease and the most common kind of dementia. It results in the destruction of brain neurons and has severe symptoms such as memory loss and difficulty learning new things. Many lines of evidence suggest that neuroinflammation is involved in the pathology of AD [119]. Kou et al. found that luteolin treatment improved spatial learning, reduced memory deficits, and inhibited astrocyte hyperactivation and neuroinflammation (TNF-α, IL-1β, IL-6, NO, COX-2, and iNOS proteins) in a triple-transgenic mouse model of AD. In addition, the expression of the ER stress markers GRP78 and IRE1α was decreased in brain tissue. In an in vitro study, LPS-activated rat C6 glioma cells and BV2 microglia cells were treated with luteolin. The researchers found that luteolin effectively alleviated cognitive impairment and limited neuronal damage by inhibiting LPS-induced cell proliferation, inflammatory cytokine production, and ER stress, as shown by the marker GRP78. Thus, luteolin has potential as a treatment for neurodegenerative diseases that are associated with neuroinflammation [120,121,122]. Ahmad et al. found that luteolin might reduce oxidative stress, neuroinflammation, apoptotic cell death, amyloid production, and synaptic dysfunction in amyloid-beta (Aβ_1–42_)-injected mice by inhibiting JNK [123]. Moreover, the intranasal delivery of luteolin-loaded chitosomes or bilosomes has been shown to be a safe, effective, and noninvasive way to alleviate AD [124,125]. In combination studies, the following treatments were also shown to be valuable therapeutic strategies for AD: palmitoylethanolamide and luteolin (co-ultra PEALut); a three-compound combination of docosahexaenoic acid, luteolin, and urolithin A called D_5_L_5_U_5_; and luteolin and l-theanine [126,127,128].

### 5.2. Parkinson’s Disease

The severe movement disorder with the highest prevalence rate worldwide is PD. It is a progressive neurodegenerative illness marked by tremors and bradykinesia [129,130]. Reudhabibadh et al. revealed that luteolin inhibited 1-methyl-4-phenylpyridinium iodide-induced mitochondrial ROS-dependent oxidative stress and apoptosis in SH-SY5Y (neuroblastoma) cells [131]. In another study, Elmazoglu et al. reported that luteolin protected microglial BV2 cells from rotenone toxicity in a stimulating manner by modulating oxidative stress responses, genes associated with PD, and inflammatory pathways [132]. In addition, Brotini et al. observed that a novel ultra-micro-quantified formulation of palmitoylethanolamide with luteolin might be an effective adjunct therapy for camptothecin in PD patients. Further studies are needed to confirm that finding [133].

## 6. Anti-Inflammatory Properties of Luteolin in Other Diseases

Luteolin has also played an anti-inflammatory and protective role in other diseases, including those of the venous endothelium, diabetes, and endometriosis [134,135,136,137].

## 7. DEGs Related to Anti-Inflammatory Cytokines/Chemokines in the GEO Dataset

Bioinformatics analysis was performed as previously described [138]. The anti-inflammation-related cytokines/chemokines were identified from differentially expressed genes (DEGs) by analyzing the available transcriptomic RNA-seq data of luteolin (GSE209778, GSE181522, and GSE111412) from the Gene Expression Omnibus (GEO; https://www.ncbi.nlm.nih.gov/geo, accessed on 10 December 2022) database, accessed on 10 December 2021, as shown in Figure 3. The “limma” in R (version 4.2.2, https://www.r-project.org/, accessed on 10 December 2022) was applied to access the profile of DEGs, which had to fit the screening criteria of adjusted *p*-value < 0.05 and |log2 (foldchange)| > 1.

We analyzed the gene expression of four cytokines/chemokines reported to be associated with anti-inflammatory effects by luteolin in high-fat diet (HFD) and HFD plus luteolin groups (see Appendix A). Figure 4 only includes the results of cytokines/chemokines that showed significant expressional changes. The gene expression of Ccl21c and Csf3 was significantly decreased in the HFD plus luteolin group compared with the HFD group. As mentioned before, studies of luteolin treatment have provided strong evidence of the anti-inflammatory activity of this compound.

Next, we analyzed the gene expression of thirty cytokines/chemokines reported to be associated with anti-inflammatory effects by luteolin in lipopolysaccharide (LPS) and LPS and luteolin groups (Appendix A). Figure 5 only includes the results of cytokines/chemokines that showed significant expressional changes. The gene expression of *Tnfsf9*, *Cxcl1*, *Ifna6*, *Wnt2*, *Csf3*, *Gdf9*, and *Cxcl5* was significantly decreased in the LPS and luteolin group compared with the LPS group, providing strong evidence that luteolin can exhibit anti-inflammatory activity.

We also analyzed the gene expression of forty five cytokines/chemokines reported to be associated with the anti-inflammatory effects of luteolin in HFD and HFD and luteolin groups (Appendix A). Figure 6 and Appendix A include the results of cytokines/chemokines that showed significant changes. The gene expression of *Timp1*, *Ccl8*, *Ccl12*, *Ccl6*, *Ccl7*, *Ccl9*, *Ccl2*, *Pf4*, *Gdf3*, *Cxcl14*, *Il33*, *Wnt2*, *Ebi3*, *Ccl3*, *Tgfb1*, *Cxcl12*, *Ccl11*, *Cxcl1*, *Ccl21a*, *Bmp1*, *Ccl4*, *Ccl24*, *Nampt*, *Cmtm7*, *Il18*, *Areg*, *Gdf10*, *Clcf1*, *Tnfsf13b*, *Grn*, *Il7*, *Ccl21c*, *Gdf15*, *Edn1, Kitl, Ifng, Osm, Il1rn, Cxcl10,* and *Cklf* was significantly reduced in the HFD and luteolin group compared with the HFD group. Therefore, these results strongly implicate that luteolin can suppress inflammatory responses via the inhibition of chemokines and cytokines at the transcriptional levels.

## 8. Conclusions and Perspectives

In vitro and in vivo studies of luteolin have provided strong evidence of its anti-oxidative and anti-inflammatory activities. In this study, we have summarized in vitro and in vivo findings about luteolin in the fields of asthma, CVDs (MI and AS), tumors (lung cancer, HCC, CRC, and pancreatic cancer), renal damage, and neurodegenerative diseases (AD and PD) during the past five years (Table 1 and Figure 7).

These studies have confirmed that luteolin regulates various mediators of inflammation and signaling pathways. For instance, luteolin could combat the COVID-19/asthma comorbidity by regulating the Toll-like receptor, MAPK, and TNF signaling pathways. The protective effect of luteolin on I/R in rats is closely related to the Siti1/NLRP3/NF-κB pathway. The upregulation of miR-6809-5p by luteolin inhibited the growth of HCC cells by inactivating the Erk1/2, p38, JNK, and NF-κB/p65 signaling pathways, thereby inhibiting growth and inducing apoptosis. In CRC, luteolin induced apoptosis and autophagy through a p53-dependent signaling pathway. Luteolin attenuated HgCl_2_-induced renal damage by activating the AMPK/mTOR autophagy pathway. In combination studies, co-treatment with luteolin killed HCC cells through JNK-mediated apoptosis. Co-ultra PEALut and D_5_L_5_U_5_ can effectively treat AD (Figure 7). Luteolin also demonstrated its anti-inflammatory properties by inhibiting the expression of various proinflammatory factors, including *IL-4, IL-5, IL-6, IL-8, IL-13, IFN-γ, TNF, IL-1β,* and *COX-2*. In addition, we analyzed some available transcriptomic RNA-seq data obtained from luteolin-treated groups and models. This bioinformatic analysis allowed us to increase the number of potential cytokines/chemokines to be targeted by luteolin. In the future, we will further analyze more DEGs from other known databases and establish the system of specific genes related to inflammatory and cancerous responses.

Despite the extensive research already conducted on the anti-inflammatory properties of luteolin in numerous disorders, more research is required for its clinical application. The poor water solubility and low bioactivity of luteolin limit its clinical application, as reported previously [144,145]. To solve that problem, many methods have been studied. Xu et al. found that glycosylation can be used to improve the solubility and bioactivity of luteolin. Zhu et al. reported that folic acid-modified, ROS-responsive nanoparticles encapsulating luteolin improved the anticancer activity of luteolin to some extent. Hsieh et al. developed a luteolin-loaded self-emulsified phospholipid preconcentrate to improve the solubility, permeability and photoprotective activity of luteolin. Li et al. revealed that D-α-tocopheryl polyethylene glycol 1000 succinate (TPGS)-coated liposomes provide an effective cancer chemotherapy strategy for model drugs with poor water solubility. Miyashita et al. found that delivering luteolin through a microemulsion system can improve its oral bioavailability without affecting its metabolite profile [146,147]. These studies suggest that if the water solubility and bioactivity of luteolin are improved by some procedures, luteolin could be developed as a therapeutic drug to treat various inflammatory and cancerous diseases caused by inflammation or oxidative stress.

## Figures and Tables

**Figure 1 ijms-24-02136-f001:**
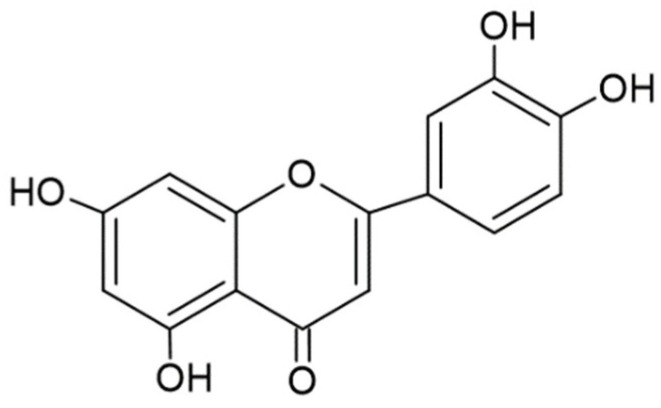
Chemical structure of luteolin.

**Figure 2 ijms-24-02136-f002:**
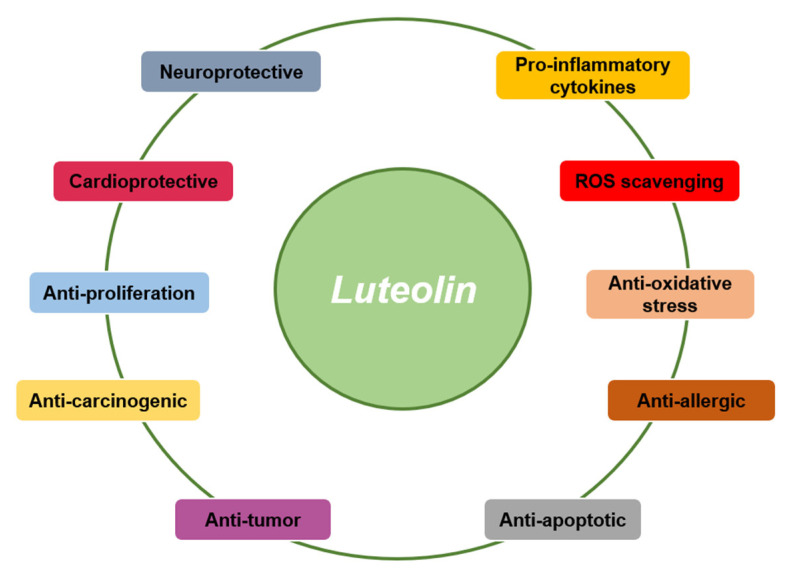
Various biological activities of luteolin. Various kinds of immunopharmacological activities of luteolin such as neuroprotective, cardioprotective, anti-oxidative, anti-allergic, and anti-tumor actions have been reported so far.

**Figure 3 ijms-24-02136-f003:**
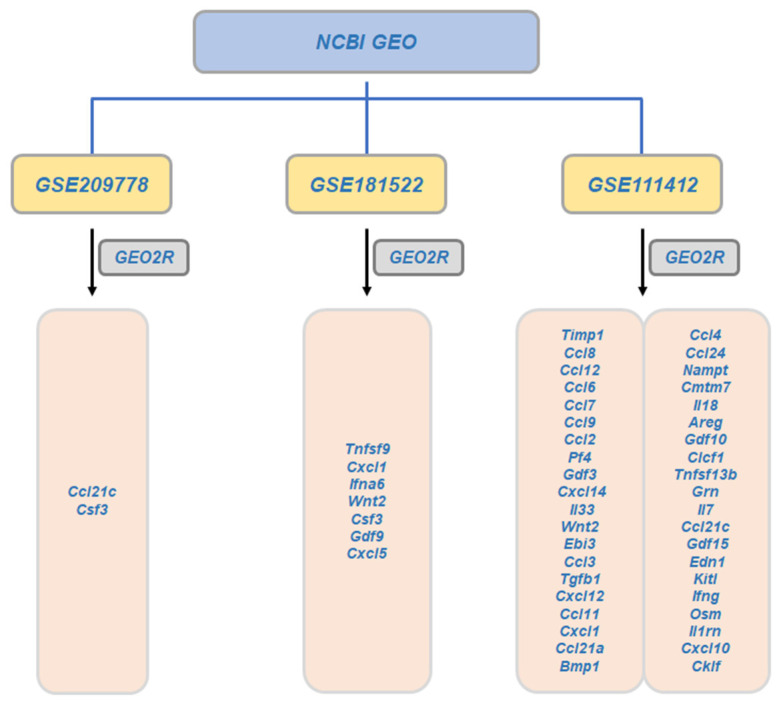
DEGs related to luteolin-treated model in the GEO dataset. Gene names that were analyzed are listed in each dataset.

**Figure 4 ijms-24-02136-f004:**
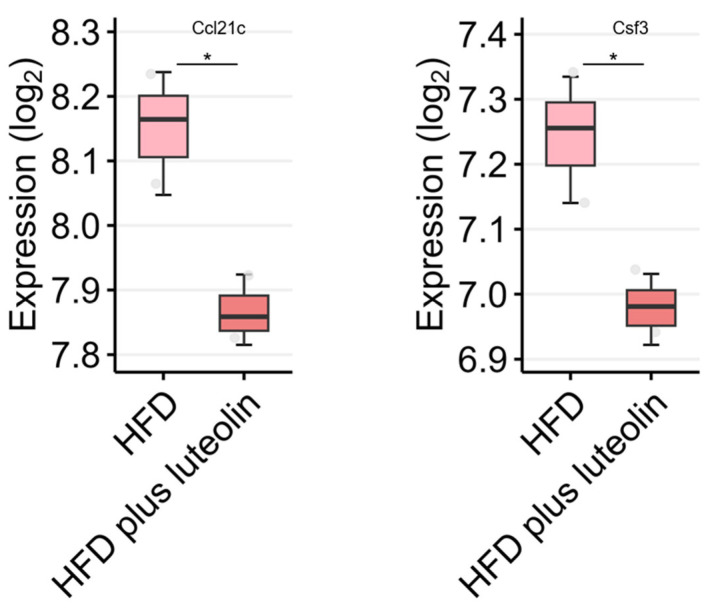
Comparison of Ccl21c and Csf3 expression in mouse muscle treated with HFD and with HFD plus luteolin. The gene expression of each cytokine/chemokine was analyzed using the GSE209778 dataset. *: *p* < 0.05 compared to the induction group.

**Figure 5 ijms-24-02136-f005:**
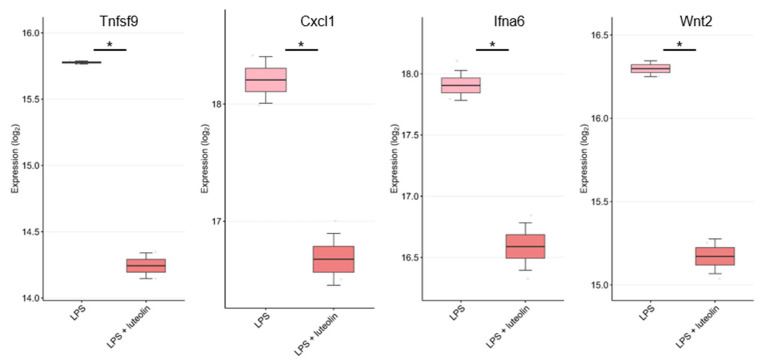
Comparison of *Tnfsf9*, *Cxcl1*, *Ifna6*, *Wnt2, Csf3, Gdf9, and Cxcl5* expression in mouse neural stem cells treated with LPS and with LPS and luteolin. The gene expression of each cytokine/chemokine was analyzed using the GSE181522 dataset. *: *p* < 0.05 compared to the induction group.

**Figure 6 ijms-24-02136-f006:**
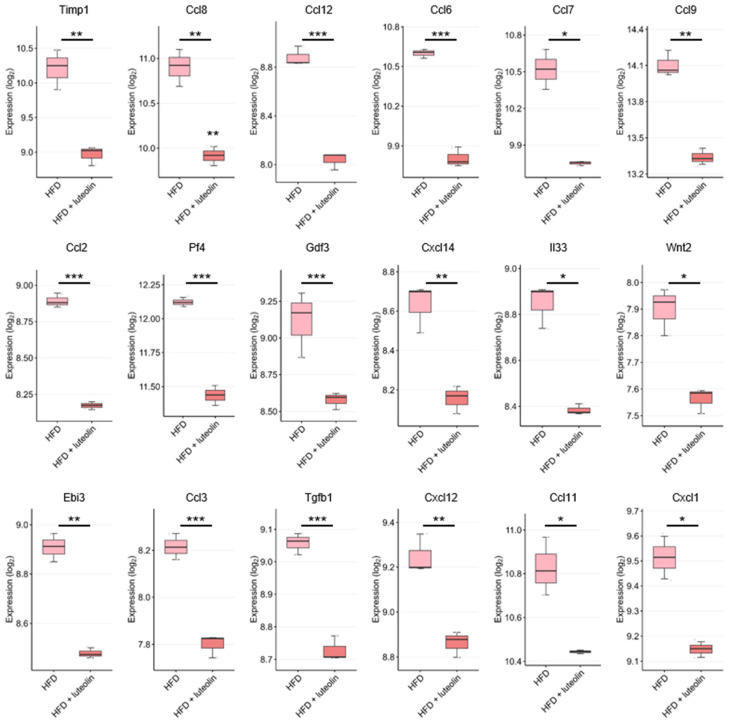
Comparison of *Timp1, Ccl8, Ccl12, Ccl6, Ccl7, Ccl9, Ccl2, Pf4, Gdf3, Cxcl14, Il33, Wnt2, Ebi3, Ccl3, Tgfb1, Cxcl12, Ccl11, Cxcl1, Ccl21a, Bmp1, Ccl4, Ccl24, Nampt, Cmtm7, Il18, Areg, Gdf10, Clcf1, Tnfsf13b, Grn, Il7, Ccl21c, Gdf15, Edn1, Kitl, Ifng, Osm, Il1rn, Cxcl10*, and *Cklf* expression in mouse epididymal adipose tissue treated with HFD and with HFD and luteolin. The gene expression of each cytokine/chemokine was analyzed using the GSE111412 dataset. ***: *p* < 0.001, **: *p* < 0.01, and *: *p* < 0.05 compared to the induction group.

**Figure 7 ijms-24-02136-f007:**
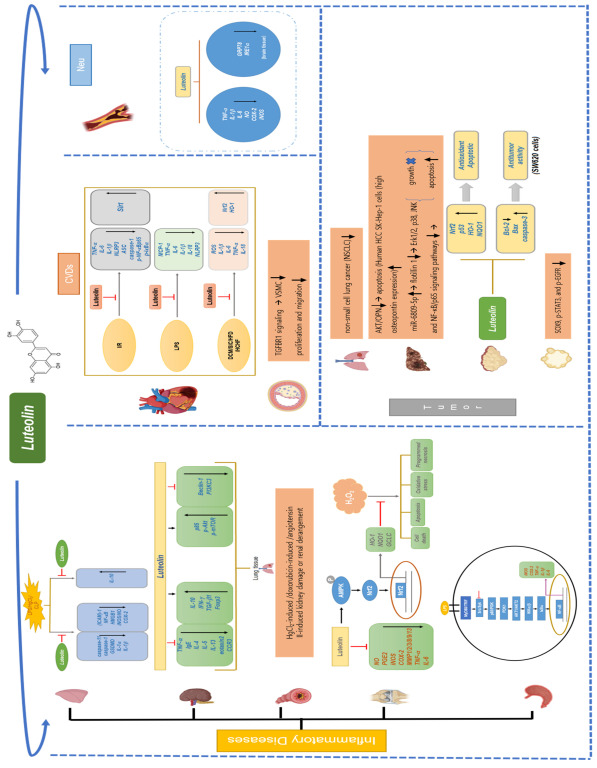
Molecular mechanism of the anti-inflammatory effects of luteolin in different diseases. The immunopharmacological effect of luteolin is summarized in terms of different tissues or organs, target molecules, stimulation conditions, and target cytokines/chemokines. Arrows in up: increase, Arrows in down: decrease, ⊥: inhibition.

**Table 1 ijms-24-02136-t001:** The mechanism of luteolin in various diseases.

Disease Type	Disease	Mechanism	Test Type	Refs.
Inflammatory Diseases	Gastritis	Luteolin changes the expression of the MUC1 extracellular domain, sT antigen, ADAM-17, IL-8, IL-10, and NF-κB	in vitro	[23]
		Luteolin eases gastritis by targeting Src and Syk in the NF-κB signaling pathway Luteolin reduces the expression of inflammatory genes (*iNOS* and *COX-2*) and pro-inflammatory cytokines (*TNF-α, IL-1β,* and *IL-6*)	in vitro and in vivo	[24,25,139,140,141,142]
	Arthritis	By reducing the expression of *JNK* and *p38MAPK* in OA cartilage cells as well as NO, TNF-α, and IL-6 By stimulating the AMPK/Nrf2 signaling pathway	in vitro	[27,28]
		Luteolin might be able to protect chondrocytes by preventing NF-κB activation	in vitro and in vivo	[20]
	Asthma	Luteolin plays a protective role in asthma by upregulating the expression of hsa_circ_0001326 By reducing the expression of *IL-1β, IL-6, the MAPK family, TNF-α, STAT3,* and *IL-17* By reducing vascular inflammation brought on NF-κB and TNF-α activation → luteolin can enhance vascular circulation	in vitro	[43,143]
		By triggering PI3K/Akt/mTOR signaling and blocking the beclin-1-PI3KC3 complex → luteolin suppresses autophagy in allergic asthma	in vitro and in vivo	[33,36]
	Renal damage	By stimulating the AMPK/mTOR autophagy pathway and Nrf2-mediated signaling → luteolin reduces HgCl_2_-induced kidney damage By lowering oxidative and inflammatory stress → luteolin inhibits apoptosis Luteolin reduces angiotensin II-induced kidney damage	in vivo	[48,49,50,51]
	Lung injury	By stimulating ERK1/2- and Ca2^+^-dependent HO-1 production, which lowers LPS-induced HMGB1, *iNOS*/NO, and *COX-2* expression → lessens endotoxin-induced ALI By inhibiting the expression of microRNA-132 → reduces LPS-induced bronchopneumonia damage Luteolin controls regulatory T cell differentiation and activates macrophage polarization against ALI	in vitro and in vivo	[57,58]
		Luteolin reduces ALI in a sepsis-related mouse model Luteolin protects lungs against damage brought on by mercuric chloride	in vitro	[53,55,56]
Cardiovascular Diseases (CVDs)	Myocardial Infarction (MI)	Modulates Nlrp3 → protect cardiomyocyte cells from LPS-induced apoptosis and inflammatory damage	in vitro	[64]
		By preventing NF-κB-mediated inflammation and triggering Nrf2-mediated antioxidant responses → defends against diabetic cardiomyopathy Through the Siti1/NLRP3/NF-κB pathway → reduces MI reperfusion damage By modulating SERCA2a via Sp1 overexpression → reduces I/R injury SHP-1/STAT3 interactions → protects against MI By promoting autophagy → reduces the cardiac damage caused by sepsis Reduces the myocardial inflammatory response brought on by a high-carbohydrate, high-fat diet.	in vivo	[67,68,69,71,73]
	Atherosclerosis (AS)	By altering the inflammatory response mediated by STAT3 → reduces AS Inhibits AMPK-SIRT1 signaling in macrophages → to reduce AS	in vivo	[78,79]
Tumors	Lung cancer	Luteolin-mediated reduction of AIM2 → might help in the treatment of NSCLC By targeting LIMK1 → suppresses the growth of lung cancer	in vitro and in vivo	[84,85]
		By blocking the focal adhesion kinase and non-receptor tyrosine kinase signaling pathways → inhibits lung cancer cells’ migration and invasion By boosting DR5 expression and Drp1-mediated mitochondrial fission → increases TRAIL sensitivity in NSCLC cells	in vitro	[86,87,88]
	Hepatocellular carcinoma (HCC)	Luteolin inhibits the AKT/osteopontin pathway → cause caspase-dependent apoptosis in SK-Hep-1 cells Luteolin controls apoptosis and autophagy in Hep3B cells Luteolin promotes the expression of miR-6809-5p, which inhibits the proliferation of HCC	in vitro	[90,91,92]
	Colorectal Cancer (CRC)	By elevating *Nrf2* expression → luteolin causes apoptotic cell death Luteolin epigenetically stimulates the Nrf2 pathway and prevents cell transformation Luteolin triggers apoptosis and autophagy via a p53-dependent pathway Luteolin causes apoptosis after oxaliplatin-induced cell cycle arrest at G0/G1 By altering the mitogen-activated protein kinase pathway → influences DNA repair By the ERK/FOXO3a signaling pathway → exhibits antitumor action that is mediated By controlling the miR384/pleiotrophin axis → prevents CRC from metastasizing	in vitro	[95,96,97,98,99,100,101]
	Pancreatic cancer	Luteolin inhibits PANC-1 cell growth and sensitizes cells to TRAIL Luteolin prevents the growth, proliferation, and epithelial–mesenchymal transition of acinar cells	in vitro	[106,107]
		STAT3 interacts with DPYD By targeting BCL-2 → suppresses the growth of pancreatic cancer	in vivo	[104,105]
Neurodegenerative disorders	Alzheimer’s disease (AD)	Luteolin reduces cognitive impairments and suppresses microglial neuroinflammatory responses	in vitro and in vivo	[121]
		By preventing ER stress-dependent neuroinflammation → reduces cognitive impairment in AD mouse models Luteolin has potential neuroprotective effects against Aβ1–42-induced AD The use of intranasal luteolin-loaded bilosomes as a brain-targeted delivery method → improved memory disorders in a mouse model of sporadic AD	in vivo	[120,123,125]
	Parkinson’s disease (PD)	Via the Erk1/2/Drp1 and Fak/Akt/GSK3 pathways	in vitro	[131,132]

## Data Availability

The data are contained within the article.

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
