# Peer review of "Immunopharmacological Activities of Luteolin in Chronic Diseases"

_ijms, 2023, doi:10.3390/ijms24032136_

Round 1
Reviewer 1 Report (Previous Reviewer 2)
The manuscript ijms-2144615 devoted the actual field of medicinal chemistry and pharmacology, namely immunopharmacological activities of Luteolin and can be interested to the specialists working in this field. The author’s opinion is clear and based on a good literature data. I am personally impressed by the structure of the article, the systematization of scientific data and the sequence of its presentation. The paper fit the Journal scope and formal requirements. However, it needs minor revision before publication.
To improve the quality and perception of the manuscript I would suggest paying attention to following comment.
A key question for improving peer-reviewed work is the following. The work undoubtedly has elements of a strategic approach to the study of flavonoids. However, flavonoids are often interpreted as PAINS (pan assay interference compounds) in the modern drug discovery process, so authors should definitely discuss this issue. In the discussion on the given question, it is worth quoting key articles on the PAINS problem. For instance - Baell J, Walters MA (September 2014). Chemistry: Chemical con artists foil drug discovery. Nature. 513 (7519): 481–3. doi:10.1038/513481a; Dahlin JL, Walters MA (July 2014). The essential roles of chemistry in high-throughput screening triage". Future Medicinal Chemistry. 6 (11): 1265–90. doi:10.4155/fmc.14.60; Baell JB, Holloway GA (2010). New substructure filters for removal of pan assay interference compounds (PAINS) from screening libraries and for their exclusion in bioassays. Journal of Medicinal Chemistry. 53 (7): 2719–40. doi:10.1021/jm901137j; etc. It would be interesting to hear the opinion of the authors on this debatable issue.
My decision is minor revision.
Author Response
Reviewer #1
- The manuscript ijms-2144615 devoted the actual field of medicinal chemistry and pharmacology, namely immunopharmacological activities of Luteolin and can be interested to the specialists working in this field. The author’s opinion is clear and based on a good literature data. I am personally impressed by the structure of the article, the systematization of scientific data and the sequence of its presentation. The paper fit the Journal scope and formal requirements. However, it needs minor revision before publication.
To improve the quality and perception of the manuscript I would suggest paying attention to following comment.
A key question for improving peer-reviewed work is the following. The work undoubtedly has elements of a strategic approach to the study of flavonoids. However, flavonoids are often interpreted as PAINS (pan assay interference compounds) in the modern drug discovery process, so authors should definitely discuss this issue. In the discussion on the given question, it is worth quoting key articles on the PAINS problem. For instance - Baell J, Walters MA (September 2014). Chemistry: Chemical con artists foil drug discovery. Nature. 513 (7519): 481–3. doi:10.1038/513481a; Dahlin JL, Walters MA (July 2014). The essential roles of chemistry in high-throughput screening triage". Future Medicinal Chemistry. 6 (11): 1265–90. doi:10.4155/fmc.14.60; Baell JB, Holloway GA (2010). New substructure filters for removal of pan assay interference compounds (PAINS) from screening libraries and for their exclusion in bioassays. Journal of Medicinal Chemistry. 53 (7): 2719–40. doi:10.1021/jm901137j; etc. It would be interesting to hear the opinion of the authors on this debatable issue.
My decision is minor revision
***Answer: Thanks for your comment. This is very good point. As suggested, we have added some additional paragraphs about PAINS (Please see line 591 to 601) with references mentioned by you. One thing more, previously, we have checked the binding site of luteolin on the ATP binding site of Src, a tyrosine kinase, with its mutants at ATP binding sites. We found that binding of luteolin is very specific, and the binding site of ATP-binding domain was conserved between other kinases. This could be the reason why luteolin has a lot of different targets and shows variety of pharmacological activities. In terms of this, we have added few sentences as well (Please see line 601 to 606).
Reviewer 2 Report (Previous Reviewer 1)
1. The review lacks novelty. There have been several reviews(over 50 published papers ) of Luteolin, what is the innovation of this review? I don't think the revised manuscript is any more innovative.
In particular, the following recently published papers:
(1)Recent Updates on Source, Biosynthesis, and Therapeutic Potential of Natural Flavonoid Luteolin: A Review. Metabolites. 2022 Nov 20;12(11):1145. doi: 10.3390/metabo12111145. PMID: 36422285; PMCID: PMC9696498.
(2)Luteolin: a flavonoid with a multifaceted anticancer potential. Cancer Cell Int. 2022 Dec 8;22(1):386. doi: 10.1186/s12935-022-02808-3. PMID: 36482329; PMCID: PMC9730645.
(3)Luteolin, a Potent Anticancer Compound: From Chemistry to Cellular Interactions and Synergetic Perspectives. Cancers (Basel). 2022 Oct 31;14(21):5373. doi: 10.3390/cancers14215373. PMID: 36358791; PMCID: PMC9658186.
(4)Anti-Inflammatory and Active Biological Properties of the Plant-Derived Bioactive Compounds Luteolin and Luteolin 7-Glucoside. Nutrients. 2022 Mar 9;14(6):1155. doi: 10.3390/nu14061155. PMID: 35334812; PMCID: PMC8949538.
(5)Role of luteolin in overcoming Parkinson's disease. Biofactors. 2021 Mar;47(2):198-206. doi: 10.1002/biof.1706. Epub 2021 Jan 14. PMID: 33443305.
and so on.
2. The figures are too simple and aesthetically lacking.
3. The content is mostly generalities, not deep enough.
Author Response
Reviewer #2
- The review lacks novelty. There have been several reviews(over 50 published papers )of Luteolin, what is the innovation of this review? I don't think the revised manuscript is any more innovative.
In particular, the following recently published papers:
(1)Recent Updates on Source, Biosynthesis, and Therapeutic Potential of Natural Flavonoid Luteolin: A Review. Metabolites. 2022 Nov 20;12(11):1145. doi: 10.3390/metabo12111145. PMID: 36422285; PMCID: PMC9696498.
(2)Luteolin: a flavonoid with a multifaceted anticancer potential. Cancer Cell Int. 2022 Dec 8;22(1):386. doi: 10.1186/s12935-022-02808-3. PMID: 36482329; PMCID: PMC9730645.
(3)Luteolin, a Potent Anticancer Compound: From Chemistry to Cellular Interactions and Synergetic Perspectives. Cancers (Basel). 2022 Oct 31;14(21):5373. doi: 10.3390/cancers14215373. PMID: 36358791; PMCID: PMC9658186.
(4)Anti-Inflammatory and Active Biological Properties of the Plant-Derived Bioactive Compounds Luteolin and Luteolin 7-Glucoside. Nutrients. 2022 Mar 9;14(6):1155. doi: 10.3390/nu14061155. PMID: 35334812; PMCID: PMC8949538.
(5)Role of luteolin in overcoming Parkinson's disease. Biofactors. 2021 Mar;47(2):198-206. doi: 10.1002/biof.1706. Epub 2021 Jan 14. PMID: 33443305.
and so on.
***Answer: Thanks for your comment. According to these comments, we have added one section about bioinformatics analysis to make the paper more innovative (Please see line 87 to 89 and line 489 to 549). Also, our view in this review paper has been more focused on explaining in vivo anti-inflammatory activities according to different tissues and organs as well as inflammation-related diseases. We are sure our paper has been improved and becoming more innovative. Since we could have only 10 days, we were unable to more expand our bioinformatic analysis level. Therefore, if you give more time, then we can make more to be further innovative compared to other review papers you mentioned above.
- The figures are too simple and aesthetically lacking.
***Answer: Thanks for your comment. We have adjusted and improved all figures to be more aesthetical (Please see all figures).
- The content is mostly generalities, not deep enough.
***Answer: Thanks for your comment. We have added some additional paragraphs about bioinformatics analysis of known luteolin data and related figures (Please see line 489 to 549 and Figs 3-6).
Reviewer 3 Report (New Reviewer)
The manuscript included the immunopharmacological activity of luteolin in a comprehensive way. However, some of the English is akward so it would be great if the author can get some native speaker help for the examination of the English.
Major suggestions:
I would strongly suggest the author to edit the english in L42, L66-67, L81, L99 etc.
Add a legend with a description for each figure
Add the Conclusion paragraph in the end of the manuscript.
The author could add more discussions about the future development of luteolin.
Table 1 looks untidy. I would suggest the author short the texts for the Mechanism.
Minor suggestions:
L11, what kind of health benefits?
L15, add "the"
L39, what means The active compound in luteolin
L46-56, add reference
L233, in the treatment of CVDs?
L239, [70,70,71].
Author Response
Reviewer #3
The manuscript included the immunopharmacological activity of luteolin in a comprehensive way. However, some of the English is akward so it would be great if the author can get some native speaker help for the examination of the English.
Major suggestions:
I would strongly suggest the author to edit the english in L42, L66-67, L81, L99 etc.
***Answer: Thanks for your comment. As suggested, we have edited the English (Please see line 43 to 44; line 67 to 68; line 83 to 89; line 103). We have also revised others as well. Since there is an English-editing service by MDPI publisher after acceptance of a paper submitted to one of MDPI journals, we hope this is not a big problem for this paper. Otherwise, we are also welcomed to get additional service from E-World Editing company (USA) if you give us more time.
Add a legend with a description for each figure
***Answer: Thanks for your comment. We have included all legends in the figures (see L70-72, 497-498, 510-512, 526-529, 544-549, and 609-611).
Add the Conclusion paragraph in the end of the manuscript.
***Answer: Thanks for your comment. As suggested, we have added more paragraph in the conclusion section (see L551-606).
The author could add more discussions about the future development of luteolin.
***Answer: Thanks for your comment. According to these comments, we have added more discussions about the future development of luteolin paragraph (Please see line 587 to 590).
Table 1 looks untidy. I would suggest the author short the texts for the Mechanism.
***Answer: Thanks for your comment. As suggested, we have corrected the Table 1 (Please see line 558 to 560).
Minor suggestions:
L11, what kind of health benefits?
***Answer: Thanks for your comment. We have added the detailed explanation of health benefits (Please see line 12 to 13).
L15, add "the"
***Answer: Thanks for your comment. We have added the “the” (Please see line 17).
L39, what means The active compound in luteolin
***Answer: Thanks for your comment. We have corrected this sentence (Please see line 40 to 41).
L46-56, add reference
***Answer: Thanks for your comment. As suggested, we have added more references (Please see line 50, and 56).
L233, in the treatment of CVDs?
***Answer: Thanks for your comment. We have included it (see line 229)
L239, [70,70,71].
***Answer: Thanks for your comment. We have corrected it (see line 291)
The style of references should be changed. In some cases, there are from 7 to 10 sources after one sentence (for example, lines 34, 42 and 103). This is unacceptable for publications in high-rated journals. Instead such references, it would be better to make a cross-reference discussion.
***Answer: Thanks for your comment. According to these comments, we have corrected the style of references (Please see L35, 43, 50, 56, 95, 103, 235; 291, 302, 334, 340, 358, 374, 407, and 456).
Round 2
Reviewer 3 Report (New Reviewer)
The quality of the manuscript has improved but there are still some minor things to imrpove.
L12, what is the diffrence of antioxidant and anti-oxidative
Can you provide figure 7 with higher resolution?
L88, add full description of DEGs
L97, should be "unanswered questions"
The author also need to update the figure number in the text, such as L102, figure 3, L119 figure 4, etc
L493, reference 138 seems not the correct reference to cite here.
Figure 6 should split into 2 figures which one of them can put to SI.
The author can add more texts about the bioinformatic analysis in Conclusions and Discussion part.
I think the last paragraph is not suitable to be in the end. The author can delete the whole paragraph or delete L592-L601.
Author Response
Reviewer #3
The quality of the manuscript has improved but there are still some minor things to imrpove.
L12, what is the diffrence of antioxidant and anti-oxidative
***Answer: Thanks for your comment. As suggested, we have corrected the English (Please see lines 12 and 33).
Can you provide figure 7 with higher resolution?
***Answer: Thanks for your comment. According to these comments, we have adjusted Figure 7 (Please see line 591).
L88, add full description of DEGs
***Answer: Thanks for your comment. As suggested, we have added the full description of DEGs (Please see line 88: “differentially expressed genes (DEGs)”).
L97, should be "unanswered questions"
***Answer: Thanks for your comment. As suggested, we have corrected the English (Please see line 97).
The author also need to update the figure number in the text, such as L102, figure 3, L119 figure 4, etc
***Answer: Thanks for your comment. As suggested, we have deleted and updated the Figure number in the text (Please see lines 102, 119, 161, 218, 394, 459, 494, 503, 516, and 530).
L493, reference 138 seems not the correct reference to cite here.
***Answer: Thanks for your comment. As suggested, we have corrected the position of reference 138 (Please see line 490).
Figure 6 should split into 2 figures which one of them can put to SI.
***Answer: Thanks for your comment. As suggested, we have adjusted the Figure6 and corrected text (Please see line 530; line 540 to 545; and SupplementaryFigure 4).
The author can add more texts about the bioinformatic analysis in Conclusions and Discussion part.
***Answer: Thanks for your comment. As suggested, we have added more texts about the bioinformatics analysis (Please see line 568 to line 573).
I think the last paragraph is not suitable to be in the end. The author can delete the whole paragraph or delete L592-L601.
***Answer: Thanks for your comment. According to these comments, we have deleted the whole paragraph.
This manuscript is a resubmission of an earlier submission. The following is a list of the peer review reports and author responses from that submission.
Round 1
Reviewer 1 Report
1. The review lacks novelty. There have been several reviews of Luteolin. In addition to more pharmacological effects, what is the innovation of this review?
2. In particular, the title is a hodgepodge.
3. The content is mostly generalities, not deep enough.
Reviewer 2 Report
The manuscript ijms-2059884 devoted the actual field of the pharmaceutical science, namely natural luteolin in regulating inflammation-related symptoms and various diseases and can be interested to the specialists working in this field. The author’s opinion is clear and based on a good literature data. I am personally impressed by the structure of the article, the systematization of scientific data and the sequence of its presentation. The paper fit the Journal scope and formal requirements. However, it needs major revision before publication.
To improve the quality and perception of the manuscript I would suggest paying attention to following comments:
1. The style of references should be changed. In some cases, there are from 7 to 10 sources after one sentence (for example, lines 34, 42 and 103). This is unacceptable for publications in high-rated journals. Instead such references, it would be better to make a cross-reference discussion.
2. References list should be carefully checked and journal style policy should be strictly followed (DOI, etc).
3. There are some grammar and orthographical errors in the manuscript, which should be corrected
My decision is minor revision.